# Impact of Alteplase on Mortality in Critically Ill Patients with COVID-19 and Pulmonary Embolism

**DOI:** 10.3390/v15071513

**Published:** 2023-07-07

**Authors:** Oleksandr Valentynovych Oliynyk, Marta Rorat, Serhij Oleksandrovych Solyarik, Vitaliy Andrijovych Lukianchuk, Serhij Oleksandrovych Dubrov, Vitaliy Hrygorovych Guryanov, Yanina Volodymyrivna Oliynyk, Svitlana Mykolaivna Yaroslavskaya, Roman Szalast, Wojciech Barg

**Affiliations:** 1Department of Anaesthesiology and Intensive Care, Bogomolets National Medical University, 01601 Kyiv, Ukraine; 2Department of Emergency Medicine, Rzeszow University, 35-310 Rzeszow, Poland; 3Department of Forensic Medicine, Wroclaw Medical University, 50-367 Wroclaw, Poland; 4Department of Anesthesiology and Intensive Care, Kyiv City Clinical Hospital No 4, 01030 Kyiv, Ukraine; 5Department of Medical Statistics, Bogomolets National Medical University, 01601 Kyiv, Ukraine; 6Department of Civilization Diseases, University of Information Technology and Management in Rzeszow, 35-310 Rzeszow, Poland; 7Department of Internal Medicine, Pneumonology and Allergology, Wroclaw Medical University, 50-367 Wroclaw, Poland; 8Department of Human Physiology, Rzeszow University, 35-310 Rzeszow, Poland; wbarg@ur.edu.pl

**Keywords:** SARS-CoV-2, anticoagulants, thrombosis, respiratory failure

## Abstract

COVID-19 is an independent risk factor for pulmonary embolism (PE). Little is known about alteplase therapy in this patient group. A retrospective study analyzed 74 patients with PE and acute respiratory distress syndrome (ARDS) due to COVID-19 who were hospitalized in the intensive care unit in 2021. Patients with or without confirmed right heart thrombi (RHT) were treated with unfractionated heparin or alteplase. The mortality rate in patients with RHT treated with heparin was 100% compared to 37.9% and 55.2% in those treated with alteplase without RHT and alteplase with RHT, respectively. The risk of death in the alteplase group increased with delayed thrombolysis (*p* = 0.009, odds ratio (OR) = 1.73 95% CI (confidence interval) 1.14–2.62), increased D-dimer concentration (*p* = 0.02, OR = 1.43 95% CI 1.06–1.93), and decreased PaO_2_/FiO_2_ ratio (*p* = 0.001, OR = 0.56 95% CI 0.41–0.78). The receiver operating characteristic method determined that a 1-day delay in thrombolytic treatment, D-dimer concentration >5.844 mg/L, and PaO_2_/FiO_2_ <144 mmHg predicted a fatal outcome. The risk of death in patients with severe COVID-19 with ARDS and PE increases with higher D-dimer levels, decreased PaO_2_/FiO_2_, and delayed thrombolytic treatment. Thrombolysis seems to be treatment of choice in severe COVID-19 with PE and RHT. It should be carried out as soon as possible after the diagnosis is established.

## 1. Introduction

The severe acute respiratory syndrome coronavirus 2 (SARS-CoV-2) pandemic started in 2019, and as of June 2023, almost 6.9 million people have died worldwide due to the coronavirus (COVID-19) [1]. The SARS-CoV-2 spike protein binds to the angiotensin-converting enzyme 2 (ACE-2) receptor, which is a cellular doorway for SARS-CoV-2 infection. In infected host immune cells, especially in CD8+ lymphocytes T, ACE-2-mediated intracellular signaling changes the immune response and amplifies the spread of the infection [2]. In addition, occupied ACE-2 receptors result in the up-regulation of angiotensin-2 (A2) and down-regulation of angiotensin 1-7 (A1-7), leading to unopposed deleterious outcomes of A2. This likely results in a microcirculatory disorder with endothelial damage, profound inflammation, and coagulopathy, which are typical in severe clinical forms of COVID-19 [3].

Numerous comorbidities, including first and foremost some metabolic, cardiovascular, and renal diseases, are enhancing the development of SARS-CoV-2 infection and making the clinical course of COVID-19 more severe. A very ample review has been recently published by Fitero and colleagues [4]. According to the authors, obesity and dyslipidemia seem to have special significance because elevated amounts of LDL may interact with macrophages in atherosclerotic plaques and increase the expression of inflammatory genes. On the other hand, small amounts of HDL additionally disrupt the immune response. In diabetes, acute viral infection not only decompensates the disease but also diabetes itself increases the affinity of SARS-CoV-2 for cellular binding and entry, reduces viral clearance and T-cell function, and intensifies susceptibility to cytokine storm.

The role of adiponectin (APN) in the interaction of metabolic disorders and COVID-19 should be emphasized. APN is concerned with the regulation of glucose metabolism, insulin sensitivity, and fatty acid oxidation but is also critical in viral infections via regulating the immune response through the anti-inflammatory/pro-inflammatory axis. A comprehensive review focused on this issue comes from Al-Kuraishy and co-workers [5]. There is a positive feedback loop between decreased APN concentration and the development of COVID infection. This augmentation may result in a hyperinflammatory response with a cytokine storm and consequently a potentially life-threatening COVID-19 infection with acute respiratory distress syndrome (ARDS).

Autoimmune diseases are the second group of pathologies significantly worsening the course of COVID-19. The existence of autoantibodies and recurrent pro-inflammatory processes results in impaired immunological tolerance and a dysfunctional immune system. Indeed, patients with various autoimmune disorders were confirmed to have more severe COVID-19. It must be also kept in mind that like other acute viral infections, COVID-19 can precipitate autoimmune diseases and/or autoinflammatory mechanisms [4].

An increasing understanding of COVID-19’s pathomechanism and the mutual relations between SARS-CoV-2 infection and existing comorbidities results in a constant evolution of therapeutic procedures. Therapeutic attempts are focused first and foremost on causal treatment with novel antiviral agents. Indeed, in addition to the most widely used remdesivir, the clinical efficacy of other drugs, like favipiravir, umifenovir, and ribavirin, was demonstrated by some authors [6,7,8].

Lungs are especially susceptible to SARS-CoV-2 infection because the vast alveoli surface is the infection gateway and, most importantly, ACE2 receptor expression is concentrated in a small population of type II alveolar cells [9]. In addition, APN seems to have the most expressed impact on the lungs [10]. Consequently, ARDS has undoubtedly been a principal contributor to mortality in COVID-19. However, the presence of thromboembolic disease is an additional factor in worsening outcomes [11,12,13,14].

COVID-19 is unquestionably a pronounced prothrombotic condition [13,15]. COVID-19-associated coagulopathy (CAC) is characterized by minimal changes in platelet counts, normal prothrombin times, and increased D-dimer and fibrinogen levels [15]. The obvious consequence of CAC is an increased risk of pulmonary embolism (PE), and this phenomenon is confirmed by recently published health insurance claims data from both Europe and the US [16,17]. There is still debate about whether PE is an outcome of deep vein thrombosis (DVT) or rather of primary local pulmonary thrombosis resulting from endothelial damage. The main pathogenetic component of clotting disorders in COVID-19 is hypoxia. In the epithelial cells of type II alveoli affected by hypoxia, the coagulation cascade is activated and leads to micro-thrombosis. In addition to a possible direct cytopathic effect, the virus may induce a local cytokine-dependent inflammatory response [18]. This response can cause massive damage to the endothelial cells of pulmonary vessels and alveolar epithelial cells with microvascular thrombosis. Viral replication causes cell infiltration and the development of a cytokine storm, triggering coagulation cascades and leading to thrombosis of pulmonary vessels [19]. It is likely that both DVT and peripheral pulmonary clotting contribute to PE incidence in COVID-19 patients, although a vast proportion of them are not diagnosed with DVT [19,20].

Data on PE incidence in COVID-19 patients are ambiguous. A meta-analysis by Sue et al. reported PE incidence in hospitalized COVID-19 patients at 16.5% (ranging from 0.7% to 57% in the studies included) depending on the individual patient’s status [20]. In general, among intensive care unit (ICU) patients, the incidence was about double compared to non-ICU patients (24.7% vs. 10.5%). In COVID-19 patients, PE may be underdiagnosed [20]. A recently published meta-analysis of autopsy data from nine studies covering 429 COVID-19 subjects (mean age 64.0 years) demonstrated that acute PE was present in about 30% of cases and was an underlying cause of death in 19.9% of them [21]. Ackerman et al. [22] demonstrated that alveolar capillary microthrombi were nine times as prevalent in the autopsies of patients with COVID-19 compared to those with influenza.

In a general population of patients with PE, right heart thrombi (RHT) are detected in <4% of patients and in up to 18% of ICU patients [23,24]. Mobile RHT are often overlooked. Right atrial appendage thrombus is a particularly underestimated cause of PE [25]. Echocardiography is the optimal diagnostic option, preferably with 3D imaging if available. However, 2D echo also provides sufficient accuracy [26]. RHT is a direct and urgent threat to the patient’s life and is associated with high mortality. Concomitant RHT quadruples 30-day mortality in patients diagnosed with acute PE (16.7% vs. 4.1%, respectively) [25]. Similar figures 21% vs. 11% and 29% vs. 16% for 14-day and three-month mortality rates, respectively come from older data [27]. Thus, RHT require immediate and effective treatment.

Potentially available treatment methods include embolectomy with extracorporeal circulation, percutaneous embolectomy, thrombolysis, or anticoagulants. However, there is still no clear consensus on optimal management. The risks of the treatment must be considered so that the survival benefits outweigh the risks of bleeding. Thus, the choice of approach depends on the individual patient’s status and the availability of the methods.

Anticoagulant therapy is recommended for hemodynamically stable patients. In unstable patients, surgical embolectomy should be performed, if available. If not, thrombolysis is proposed [26]. Recombinant tissue plasminogen activators (rtPA) are preferred over first-generation thrombolytic agents (streptokinase and urokinase) for this treatment. Alteplase is frequently used in acute PE, while other rtPA products like reteplase, desmoteplase, or tenecteplase are being researched for treatment of this condition [23]. The efficacy and safety of alteplase in patients with severe COVID-19 were demonstrated very recently [28,29]. A large phase IIb/III study (TRISTARDS), which started in October 2021, was terminated due to sponsor decision, but no results have been published yet [30].

Little is known about RHT in patients with severe COVID-19 and PE. The literature only includes infrequent case studies with a variety of therapeutic attempts (including i.v. heparin, LMWH, warfarin, and alteplase) and diverse clinical effects [31,32,33,34,35,36]. Consequently, the prevalence and mortality rate in patients with severe COVID-19 and RHT remain unknown.

The aim of this retrospective, single-center study was to estimate the effectiveness of thrombolytic treatment with alteplase and to find out the predictors of mortality in ICU patients with COVID-19, ARDS, and PE, with or without RHT. In the literature, to our best knowledge, there are no papers focused directly on this topic.

## 2. Materials and Methods

A retrospective study analyzed the medical records of 364 patients with COVID-19 and ARDS who were hospitalized in the Department of Anesthesiology and Intensive Care for Infectious Patients, Kyiv City Clinical Hospital № 4 from 1 January 2021 to 28 December 2021. We identified 74 records meeting the following inclusion and exclusion criteria.

Inclusion criteria:SARS-CoV-2 infection confirmed with positive reverse transcription-polymerase chain reaction (PCR) test or positive rapid antigen test;Clinical symptoms of interstitial pneumonia confirmed with a high-resolution computed tomography (HR-CT) scan;Acute respiratory failure with partial pressure of oxygen in arterial blood (PaO_2_) < 60 mm Hg when breathing atmospheric air;Serum D-dimer concentration > 700 µg/L;Echocardiographic diagnosis of PE confirmed with CT pulmonary angiography (CT-PA), if possible due to the severity of the clinical condition;High-risk PE-related death as defined by ESC Guidelines [23].

Exclusion criteria:History of hemorrhagic stroke or stroke of unclear etiology;Ischemic stroke within the last 6 months;Neoplasia of the central nervous system;Polytrauma, surgery, head injury within the last 3 weeks;Hemorrhagic diathesis;Active hemorrhage;Participation in other interventional studies within 3 months of enrolment.

All patients underwent duplex ultrasound scanning with VOLUSON 730 EXPERT or VIVID 7 (GE, Tampa, FL, USA) devices using 2–5 and 4–6 MHz convex transducers and 5–12 MHz linear transducers. The echocardiographic criteria for PE were based on the measurement of tricuspid regurgitation flow velocity and right ventricular (RV) volume. Diagnosis of PE was based on either impaired RV ejection defined as a pulmonary acceleration time (PAT) less than 60 milliseconds together with a tricuspid regurgitation (TR) jet gradient less than 60 mm Hg (“60-60” sign) [37], a regional pattern of RV dysfunction with akinesia of the mid free wall together with normal apex motion (McConnell sign) [38], or both. The diagnosis of PE was confirmed via CT-PA in 67/74 (90.5%) patients. CT-PA could not be performed in the remaining 7 patients due to unstable hemodynamics. All CT examinations were performed using an AQUILION 64-slice CT scanner (Toshiba, Tokyo, Japan). In order to verify the diagnosis of PE, a post-mortem examination was performed in 40 (93%) of the deceased patients. In all cases, the diagnosis of thromboembolism has been confirmed.

The criteria for high-risk PE-related death were shock or hypotension and RV overload (dysfunction) on echo scans and/or on CT-PA. In such hemodynamically unstable patients, the Pulmonary Embolism Severity Index (PESI) need not be marked [23]. Shock or hypotension were identified if systemic systolic blood pressure was <90 mm Hg or the pressure dropped <40 mm Hg for >15 min (unless caused by arrhythmia, hypovolemia, or sepsis). Echo signs of right ventricular dysfunction were defined as the presence of >1 of 3 signs:RV diastolic volume (parasternal access) > 30 mm or RV/left ventricle (LV) ratio > 1;Interventricular systolic septal flattening;Acceleration time < 90 ms or a tricuspid regurgitation pressure gradient > 30 mm Hg in the absence of LV hypertrophy.

In general, for the treatment of patients with severe COVID-19, the Ukrainian protocol “Providing medical assistance for the treatment of the coronavirus disease (COVID-19), Ministry of Health of Ukraine, 2020” was used. The protocol provides no detailed guidelines for the treatment of PE as a COVID-19 comorbidity. Therefore, we used the 2019 ESC Guidelines for the diagnosis and management of acute pulmonary embolism [23].

Thrombolytic treatment with alteplase was proposed to all the patients, but this is not reimbursed in Ukraine. Consequently, some of the patients did not take this treatment. Patients were thus allocated to one of the following groups based on the treatment decision and on clinical and imaging data:Group I—patients without right heart thrombi (RHT), i.e., without right atrial (RA) or RV thrombus, treated with thrombolysis;Group II—patients with RHT, treated with thrombolysis;Group III—patients with RHT and treated with heparin only.

In all patients, unfractionated heparin (UFH, heparin sodium) was administered intravenously at an initial dose of 80 U/kg, followed by an infusion of 18 U/kg/h for 3 to 5 days. The UFH doses were adjusted so that the APTT was kept within 70–100 s [39]. Low-molecular-weight heparin (LMWH, enoxaparin) was administered subcutaneously at a dose of 80 anti-Xa IU/kg every 12 h following treatment with UFH.

In patients from Groups I and II, attempts to start thrombolytic treatment began as soon as possible after the indications for this treatment were established. The delay in starting treatment was due to the lack of reimbursement of alteplase in Ukraine. According to a local protocol, alteplase (Actilyse, Boehringer-Ingelheim, Ingelheim, Germany) was administered for 2 h at a total dose of 100 mg. The administration started with 10 mg as an intravenous bolus for 1–2 min followed by an infusion of 90 mg for 2 h. For patients weighing less than 65 kg, the total dose did not exceed 1.5 mg/kg of body weight. All participants were treated with methylprednisolone 32 mg or dexamethasone 6 mg daily (for 10–14 days from admission), antibiotics (in case of suspected or confirmed bacterial infection), and balanced fluid therapy.

All patients received respiratory support depending on the level of respiratory insufficiency. In the majority of patients, the respiratory index (PaO_2_/FiO_2_, i.e., the ratio of arterial oxygen partial pressure to the fraction of inspired oxygen) was <200 mm Hg. Patients with a respiratory index of 100–200 mm Hg initially received non-invasive ventilation and those with a respiratory index <100 mm Hg were intubated and received invasive pressure support ventilation.

The patients’ medical history and laboratory results were analyzed, including body mass index (BMI), complete blood count, respiratory index, ferritin, IL-6, C-reactive protein, and D-dimer concentrations. Obesity was defined as BMI > 30. Mortality was assessed over 28 days from admission to the intensive care unit (ICU). Thrombolysis delay was defined as the number of days that passed from the diagnosis of thrombosis to the start of alteplase treatment.

### Statistical Analysis

MedCalc^®^ Statistical Software version 20.112 (MedCalc Software Ltd., Ostend, Belgium; https://www.medcalc.org; 2022, accessed on 1 March 2022) was used for the analysis. For quantitative variables, the Shapiro–Wilk test was performed to test for normality. If the distribution was non-normal, the median (Me) and interquartile range (QI-QIII) were calculated, and non-parametric methods were chosen for further data analysis. The Kruskal–Wallis test was used to compare quantitative characteristics between the groups. For pairwise comparison, Dunn’s test was used. For qualitative variables, occurrence (%) was calculated. For the comparison of categorical variables, the chi-square test was used, and for pairwise comparisons of subgroups, the Bonferroni correction was performed. Logistic regression analyses and odds ratios (OR) with a 95% confidence interval (CI) were used to evaluate the effect of factors on the phenomenon being studied. The predictive accuracy of the model was assessed by the area under the curve (AUC) of the receiver operating characteristic (ROC), and its 95% CI was calculated. The *p*-value < 0.05 was considered significant for all analyses.

## 3. Results

The basic medical and demographic data of the population studied as well as 28-day invasive ventilation and mortality rates are presented in Table 1. It should be emphasized that entirely all patients with RHT who did not undergo thrombolysis (Group III) deteriorated, required mechanical ventilation, and died. Treatment with alteplase did not result in any serious complications. Of 58 patients from Groups I and II, 8 experienced epistaxis and 1 experienced hemorrhoidal bleeding. Tranexamic acid tamponade was used to stop epistaxis. Hemorrhoidal bleeding resolved without treatment.

Logistic regression was used to identify factors associated with the risk of death. As all the patients who did not undergo thrombolysis had RHT and all of them died, those two factors (thrombolytic treatment and RHT) were excluded from this analysis. Consequently, nine potential factors were analyzed: obesity; sex; age; day of PE diagnosis (since the onset of COVID-19 symptoms); D-dimer, ferritin, and CRP concentrations; white blood cell (WBC) and platelet (PLT) count; and respiratory index PaO_2_/FiO_2_ (Table 2).

A single-factor analysis showed that the risk of death increased with a patient’s age and D-dimer concentration and decreased with the PaO_2_/FiO_2_ ratio. Logistic regression modelling was used to identify factors associated with the risk of mortality in patients who underwent thrombolysis (58 patients). Eleven parameters were analyzed as potential risk factors: age, days since COVID-19 onset, days since pulmonary embolism diagnosis, obesity, D-dimer, CRP, WBC, platelets, PaO_2_/FiO_2_, ferritin, and presence of right heart thrombus. Table 3 presents the results of the mortality risk analysis in single-factor logistic regression models.

A single-factor analysis demonstrated an increased risk of death with delayed thrombolysis, increased D-dimer concentration, and decreased respiratory index. Using the receiver operating characteristic method, a model defining critical thresholds for those features was constructed. According to this model, a 1-day delay in thrombolytic treatment, D-dimer concentration > 5.844 mg/L, and PaO_2_/FiO_2_ < 144 mm Hg predicted a fatal outcome (Figure 1).

A multivariate logistic regression analysis was used to specify a set of factors increasing the risk of mortality in patients who underwent thrombolysis. Two main risk factors were identified: the delay in thrombolysis from the PE diagnosis (per 1 day) and respiratory index (PaO_2_/FiO_2_). Table 4 presents the coefficient analysis of the model.

Multivariate analysis showed an increased (*p* = 0.03) risk of death with each day of thrombolysis delay (when standardized by PaO_2_/FiO_2_). As PaO_2_/FiO_2_ increases, the risk of death decreases for every 10 mm Hg (when standardized for thrombolysis delay). Figure 2 shows the ROC curve of the two-factor model for mortality risk prediction. The area under the ROC curve demonstrated a strong correlation between the risk of death and the thrombolysis delay, as well as the respiratory index PaO_2_/FiO_2_.

## 4. Discussion

The main finding of our retrospective research was that in patients with severe COVID-19 with acute respiratory failure, pulmonary embolism (PE) with accompanying right heart thrombus (RHT) results in death if not treated with thrombolysis. Treatment with alteplase reduces the risk by nearly half. The incidence of RHT in our population was as high as 60.1% but did not appear to directly impact mortality. In a single-factor analysis, three factors correlated with mortality in all patients studied: increasing age, D-dimer concentration, and decreasing PaO_2_/FiO_2_ ratio (Table 2). In patients treated with thrombolysis, increased D-dimer concentration, decreased respiratory index, and delayed thrombolysis were found to be significant (Table 3).

In general, PE is reported to increase the risk of death in COVID-19 patients, but not all studies confirm this finding. A very early meta-analysis on 1835 COVID-19 patients reported the incidence and mortality rate in those not developing and those developing PE as 15.3% and 45.1%, respectively [40]. Mir et al. [41] analyzed data from 439 ICU patients with COVID-19 and reported that in-hospital mortality was comparable between PE and non-PE patients. Another meta-analysis comprising 16 trials and 5826 non-ICU COVID-19 patients reported a pooled prevalence of PE of 32% when diagnosed with computed tomography pulmonary angiography. Significant correlations between PE and the male gender, mechanical ventilation, intensive care unit admission, and circulating D-dimer and CRP concentrations were found, but there was no significant correlation between PE and mortality [42]. In contrast, a meta-analysis from Italy covering 1681 patients from eight trials demonstrated acute PE present in 19.0% of them. These patients were also at a higher risk of mortality compared with those without PE [43].

Data from literature on RHT in COVID-19 patients is infrequent and mainly relates to case reports. The majority of them reported successful treatment with low-weight heparin in hemodynamically stable patients with COVID-19 and RHT [31,32,34,36,44], but there are also reports of more severe cases. Carrizales-Sepúlveda et al. [33] reported a case of a 62-year-old male who was diagnosed with circulatory shock and RHT on the 13th day of his COVID-19 infection; he was subsequently treated with alteplase. Another report relates to a 56-year-old male with multiple thrombi of the right atrium and right ventricle, severe dilatation of the right cavities with right ventricle overload, and significant pulmonary arterial hypertension (PAH) of 110 mm Hg, in whom thrombolysis with reteplase was performed [45]. Another 56-year-old man developed severe PAH and was diagnosed with a large free-floating thrombus in the right atrium protruding through the tricuspid valve on his 10th day of COVID-19 infection. The patient was treated with rtPA (the report does not specify the name) [46]. A case of a 51-year-old diabetic male with ARDS, PE, and massive RHT in whom thrombolysis with alteplase was performed was also reported [47]. Thrombolysis was performed successfully in all the patients mentioned, and neither death nor serious complications were reported. This literature data cannot be directly compared with ours. Our research only concerned critically ill COVID-19 patients with ARDS and PE; thus, the patients’ status most likely resulted in such a high incidence of RHT.

Our retrospective population comprised patients with PE with or without RHT who underwent thrombolysis, but among those who did not undergo this treatment, only patients with PE and RHT were present. All patients not receiving thrombolytic treatment died. Of course, the lack of thrombolytic treatment was not necessarily the only factor influencing mortality. Group III patients were significantly older and had significantly higher CRP levels as compared to the ones in Group I (Table 1). In addition, logistic regression analysis found age to be a significant risk factor for death (Table 3). Anyway, thrombolysis seems to be the necessary condition for surviving in critically ill COVID-19 patients with ARDS, PE, and RHT. At our center, we were not able to perform surgical embolectomy or catheter-based thrombolysis. Thus, only systemic thrombolysis was used to give our patients a chance to survive.

The risk factors of death in patients with PE identified in our analysis—age and D-dimer concentration and decreasing PaO_2_/FiO_2_ ratio—are consistent with those from the literature. Kutsogiannis et al. [48] demonstrated in a cohort of 85 COVID-19 or non-COVID-19 patients with ARDS that independent predictors of 60-day mortality included increasing ventilatory ratio, IL-6, D-dimer, and decreasing respiratory index. A meta-analysis comprising 13,310 COVID-19 patients showed hemoglobin and D-dimer levels were associated with higher mortality in multivariable logistic regression analysis [49]. A recently published analysis of 549 COVID and 439 non-COVID patients with PE revealed a higher risk of in-hospital death in COVID patients (12.8% vs. 5.3%, *p* < 0.001). The severity of PE assessed according to right ventricular dysfunction and PESI score were independently associated with in-hospital mortality in COVID patients [50]. Because thrombolysis was demonstrated to be a prerequisite, we also analyzed risk factors for death in those who received this treatment. Again, in a single-factor analysis, increased D-dimer concentration and decreased respiratory index, as well as delayed thrombolysis, were found to increase the risk of death. Using the ROC approach, we were able to find the cut-off values for those factors. A one-day delay in thrombolysis treatment, D-dimer concentration >5.844 mg/L, and PaO_2_/FiO_2_ <144 mm Hg were predictors of death in this model (Figure 1). With a multivariate logistic regression, two risk factors were identified, namely time from diagnosis of PE to alteplase administration and respiratory index predicting death with high accuracy (Table 4 and Figure 2). It is noteworthy that the risk of death increased by 1.75 times per day of delay (if standardized for respiratory index). Thus, our data strongly suggest that if thrombolysis is indicated, it should be carried out as soon as possible. Due to the uniqueness of our analysis, it is not possible to compare the results obtained with those from other studies.

We attempted to confirm a hypothesis that RHT increased mortality risk in our patients. Right heart thrombus was diagnosed in as many as 45 of 74 (60.1%) of the patients examined. This surprisingly high percentage of patients with RHT was likely a result of the inclusion criteria used. Only ICU patients with PE and ARDS were enrolled in the study. In the RHT subgroups, 32 (71.1%) died, including 16 of 29 (55.2%) who received alteplase and all 16 who refused this treatment. Thus, we could only directly compare the impact of RHT on mortality in those who received thrombolysis. As presented in Table 1, mortality among patients with or without RHT was 55.2% and 37.9%, respectively, but the difference was not statistically significant at *p* = 0.88 (not shown). In addition, neither single-factor nor multifactor regression analyses confirmed the influence of RHT on mortality (Table 2 and Table 3). Consequently, we cannot claim that in COVID-19 patients with ARDS and PE, RHT is an independent risk factor for death.

There is an open question about a possible outcome in patients with severe COVID-19 and PE but without RHT, who would not be treated with alteplase. There were no such patients in our study. Studies addressing the thrombolytic treatment of PE in COVID-19 patients are infrequent, but the majority of them reported a successful outcome with this procedure. Philippe et al. [51] presented a case series of seven patients diagnosed with PE who were treated with systemic thrombolysis. Clinical improvement was obtained in five of them, including an improvement in respiratory index and a decrease in RV dysfunction. Another series of six patients were coming from India. The patients were diagnosed with COVID-19, ADRS, and PE, and thrombolytic therapy was successfully performed in all of them [52]. Betancourt-Del Campo et al. [53] presented a case of a 69-year-old man with severe COVID-19, respiratory failure, and acute right ventricular insufficiency who was successfully treated with alteplase. Aguilar-Piedras et al. [54] presented an interesting case of a 53-year-old multimorbid patient with COVID-19, who rapidly deteriorated due to pneumothorax and PE and was successfully treated with a half-dose of alteplase. There were also recent reports of successful thrombolysis in geriatric patients [55,56]. In contrast, So et al. presented a cohort of 57 COVID-19 patients who received tPA for PE. Improvement following the treatment was reported in 49.1% (28/57) of them, but only six patients (10.5%) survived to discharge [57]. Because in our research, RHT did not increase the risk of death, we can only speculate that without thrombolysis, such patients would have most likely died as well. Thus, thrombolysis seems to be a mandatory treatment in this group, too.

Our research has several limitations. Firstly, this is a retrospective and single-center study. Secondly, it uses a relatively small number of patients. However, these patients form a fairly homogeneous group as defined by inclusion and exclusion criteria. Consequently, the conclusions can be applied to critically ill ICU patients with COVID-19 but cannot be extrapolated to other COVID-19 patients. There is a need for prospective studies evaluating the effectiveness of thrombolysis in COVID-19 patients with varying degrees of thrombosis.

## 5. Conclusions

The risk of death in patients with severe COVID-19 with ARDS and pulmonary embolism increases with a 1-day delay in thrombolytic treatment, D-dimer concentration > 5.844 mg/L, and PaO_2_/FiO_2_ < 144 mm Hg. Right heart thrombus is not an independent mortality risk factor in those patients. Thrombolysis seems to be the treatment of choice in severe COVID-19 patients with PE and RHT. It should be carried out as soon as possible after the diagnosis is established. Further prospective and multicenter trials are needed to verify the results.

## Figures and Tables

**Figure 1 viruses-15-01513-f001:**
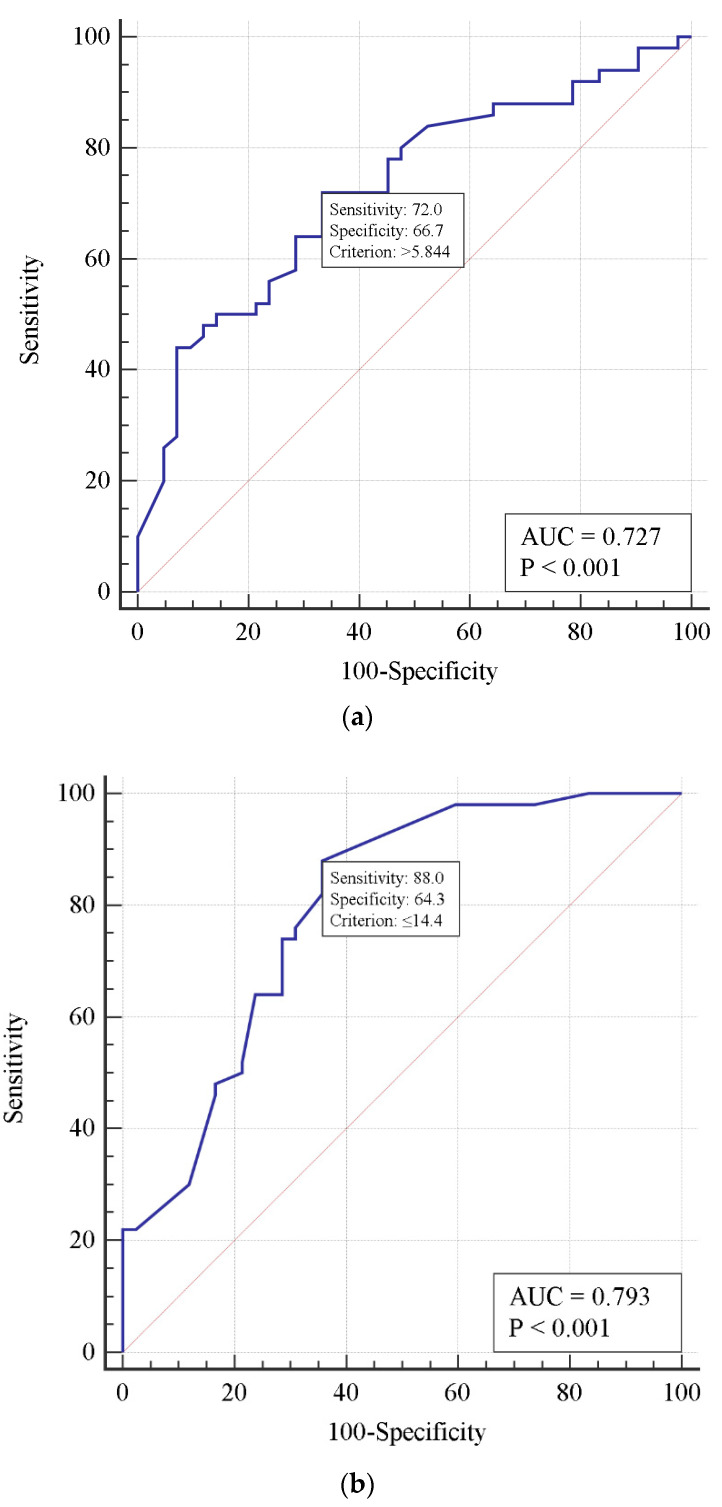
Receiver operating characteristic curves predicting risk of mortality in patients who underwent thrombolysis for D-dimer concentration (**a**), PaO_2_/FiO_2_ ratio (**b**), and a 1-day thrombolysis delay (**c**).

**Figure 2 viruses-15-01513-f002:**
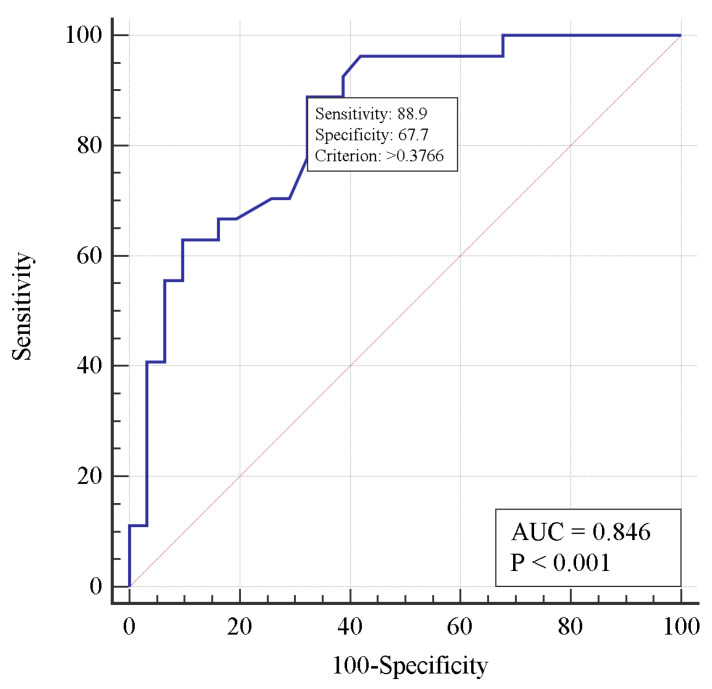
Receiver operating characteristic curve for predicting mortality risk in a two-factor logistic regression model.

**Table 1 viruses-15-01513-t001:** Basic demographic and medical data, and invasive ventilation and mortality rates in the studied cohort. Median (QI–QIII) for continuous variables, numerical values (%) for categorical variables.

	Group I (*n* = 29)	Group II (*n* = 29)	Group III (*n* = 16)	*p* Value
Age, years	62 (55.5–68.25) ^c^	66 (58–67.25)	71 (64–80) ^a^	0.02
Sex, female (%)	12 (41.4%)	12 (41.4%)	9 (56.%)	0.57
Obesity	21 (72.4%)	21 (72.4%)	14 (87.5%)	0.46
BMI, kg/m^2^	32 (29–34)	33 (29–34)	32 (31–32.5)	0.51
D-dimer, mg/L	5.66 (5.32–8.79)	6.4 (5.44–8.98)	5.98 (5.66–9.70)	0.09
CRP, mg/L	52 (39–73) ^c^	56 (48.75–61.5) ^c^	72 (61–80) ^a^	0.01
Leukocytes, ×10^9^/L	4.2 (4.150–5.4)	4.4 (4.2–5.425)	4.75 (4.2–5.5)	0.24
Thrombocytes, ×10^9^/L	128 (127.5–146)	136 (128–144.25)	136 (128–142)	0.84
PaO_2_/FiO_2_	132 (113.8–176) ^b^	112 (99–132.25) ^a^	125.5 (117.5–139)	0.02
Ferritin, mg/L	1.04 (0.74–1.28)	0.89 (0.56–1.46)	1.13 (1.08–1.25)	0.40
Duration of COVID-19 symptoms before hospital admission (days)	18 (12.25–20)	17 (12.25–20)	19 (15.5–23.5)	0.36
Artificial ventilation	15 (51.7%) ^c^	16 (55.2%) ^c^	16 (100%)	0.003
Death	11 (37.9%) ^c^	16 (55.2%) ^c^	16 (100%)	<0.001

Group I—patients without right atrial (RA) or right ventricular (RV) thrombus; Group II—patients with RA or RV thrombus; Group III—patients with RA or RV thrombus without thrombolytic treatment. The Kruskal–Wallis test was used in the comparison; subsequent comparisons were performed with Dunn’s test. A statistically significant difference (*p* < 0.05): ^a^ from Group I; ^b^ from Group II; ^c^ from Group III.

**Table 2 viruses-15-01513-t002:** Analysis of mortality risk in single-factor logistic regression models.

Factor		The Model Coefficient, b ± m	Significance Level of Difference of the Coefficient from 0, *p* Value	Odds Ratio, OR (95% CI)
Obesity	no	Reference		
	yes	0.44 ± 0.55	0.43	–
Sex	female	Reference		
	male	0.26 ± 0.47	0.58	–
Age, per year		0.059 ± 0.022	0.007	1.07 (1.02–1.11)
CRP, per mg/L		0.03 ± 0.14	0.82	–
D-dimer, per mg/L		0.34 ± 0.14	0.015	1.40 (1.07–1.85)
Leukocytes, per 10^9^/L		−0.28 ± 0.21	0.19	–
Thrombocytes, per 10^10^/L		−0.08 ± 0.13	0.54	–
Ferritin, per mg/L		0.37 ± 0.65	0.57	–
PaO_2_/FiO_2_, per 10 mm Hg		−0.49 ± 0.13	<0.001	0.62 (0.48–0.79)

**Table 3 viruses-15-01513-t003:** Analysis of risk of death in a single-factor logistic regression model in patients who underwent thrombolysis.

Factor		The Model Coefficient, b ± m	Significance Level of Difference of the Coefficient from 0, *p* Value	Odds Ratio, OR (95% CI)
Obesity	no	Reference		
	yes	0.16 ± 0.59	0.79	–
Age, per year		0.042 ± 0,024	0.078	–
CRP, per mg/L		−0.21 ± 0.17	0.20	–
D-dimer, per mg/L		0.36 ± 0.15	0.02	1.43 (1.06–1.93)
Leukocytes, per 10^9^/L		−0.56 ± 0.30	0.06	–
Thrombocytes, per 10^10^/L		−0.05 ± 0.13	0.71	–
Ferritin, per mg/L		−0.05 ± 0.69	0.94	–
PaO_2_/FiO_2_, per 10 mm Hg		−0.57 ± 0.17	0.001	0.56 (0.41–0.78)
RHT	no	Reference		
	yes	0.70 ± 0.53	0.19	–
Thrombolysis delay, per one day		0.55 ± 0.21	0.009	1.73 (1.14–2.62)

RHT—right heart thrombus.

**Table 4 viruses-15-01513-t004:** Two-factor logistic regression model analysis of mortality risk in patients who underwent thrombolysis.

Factor	The Model Coefficient, b ± m	Significance Level of Difference of the Coefficient from 0, *p* Value	Odds Ratio, OR (95% CI)
Thrombolysis delay, per one day	0.56 ± 0.26	0.03	1.75 (1.04–2.93)
PaO_2_/FiO_2_, for 10 mm Hg	−0.55 ± 0.17	0.001	0.58 (0.42–0.80)

## Data Availability

The data presented in this study are available on request from the corresponding author. The data are not publicly available due to ethical aspects.

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
