# Peer review of "Impact of Alteplase on Mortality in Critically Ill Patients with COVID-19 and Pulmonary Embolism"

_viruses, 2023, doi:10.3390/v15071513_

Round 1
Reviewer 1 Report
The present article assesses the efficacy of alteplase for pulmonary embolism in critically ill COVID-19 patients by using retrospective data. The topic is relevant and up-to-date, but certain deficiencies identified in both content and form need to be addressed based on the specific recommendations below:
No full stop at the end of a title.
Please see the instructions for the authors and the template provided by the journal in order to reorganize the author section accordingly (ID is not needed).
The conclusion part of the abstract should be improved in terms of outcomes and what future research directions this research may refer to.
L72-95 the information is organized in the form of an overly long paragraph, which decreases readability and comprehension. Please reorganize into shorter paragraphs that will be more logical and easier to understand.
L96- "The literature only includes infrequent case studies". A short description of the conclusions of these studies should be included.
The aim of the paper should be presented separately in the last paragraph of the introduction section and improved from the perspective of describing the contribution to the field under analysis and the elements of scientific novelty presented.
The general management within COVID-19 should be briefly mentioned in order to provide the current image. It would be advisable to discuss the pathophysiological mechanism in order to understand from where the pulmonary issues may arise, to present the most relevant comorbidities and associated pathologies, some proteins with diagnostic and prognostic roles, and last but not least, the treatment. I suggest checking and referring to the following updated sources: PMID: 36406478; PMID: 35131656; PMID: 36211634.
Heparin and alteplase have different mechanisms of action and effects. The difference between the two should be better discussed, and whether there was a well-designed protocol for which molecules should be used depending on the medical context.
Conclusions are too poorly presented in relation to the complexity of the topic addressed.
Author Response
Dear Reviewer, thank you very much for the review and valuable comments.
Please find the answers to your questions below:
1. No full stop at the end of a title.
It is changed
2. Please see the instructions for the authors and the template provided by the journal in order to reorganize the author section accordingly (ID is not needed).
It is changed
3. The conclusion part of the abstract should be improved in terms of outcomes and what future research directions this research may refer to.
It is changed
4. L72-95 the information is organized in the form of an overly long paragraph, which decreases readability and comprehension. Please reorganize into shorter paragraphs that will be more logical and easier to understand.
It is changed
5. L96- "The literature only includes infrequent case studies". A short description of the conclusions of these studies should be included.
Please note that this was talked over in the Discussion. A brief summary was added now in the Introduction section (lines: 98-99).
6. The aim of the paper should be presented separately in the last paragraph of the introduction section and improved from the perspective of describing the contribution to the field under analysis and the elements of scientific novelty presented.
It is changed (lines: 102-105)
7. The general management within COVID-19 should be briefly mentioned in order to provide the current image. It would be advisable to discuss the pathophysiological mechanism in order to understand from where the pulmonary issues may arise, to present the most relevant comorbidities and associated pathologies, some proteins with diagnostic and prognostic roles, and last but not least, the treatment. I suggest checking and referring to the following updated sources: PMID: 36406478; PMID: 35131656; PMID: 36211634.
Indeed, discussing the pathophysiological mechanism of COVID-19 including molecular mechanisms with their diagnostic and prognostic roles and with a special address to pulmonary pathologies, as well as presenting the most relevant comorbidities and associated pathologies together with current treatment options would be a very good background for this paper. However, this is not a reviewing paper but a retrospective clinical trial focused on a very narrow issue of COVID-19 and one comorbidity and the efficacy of its treatment. We are afraid that the proposed extension is far above for a paper of this kind.
8. Heparin and alteplase have different mechanisms of action and effects. The difference between the two should be better discussed, and whether there was a well-designed protocol for which molecules should be used depending on the medical context.
Information about the protocol for COVID-19 treatment applicable in Ukraine was added to the Materials & Methods section (lines: 161-166).
9. Conclusions are too poorly presented in relation to the complexity of the topic addressed.
It is changed
Reviewer 2 Report
The authors analyzed the efficacy of alteplase in critically ill COVID19 patients with ARDS and pulmonary embolism. All parts of the manuscript are well written. The introduction section provides a very comprehensive explanation about the pathophysiology of thrombosis and pulmonary embolism in COVID 19 patients. Although there was a small number of patients in the Group III (RHT without thrombolysis), the very interesting finding is that all of them had died. This finding should be further analyzed with a larger number of patients.
The other interesting finding is that the risk of death was increased with each day of thrombolysis delay. After the diagnosis of pulmonary embolism was made, when was thrombolysis administered? What was the reason for the later administration of thrombolysis in some patients?
I have no further comments.
Thank you
Author Response
Dear Reviewer, thank you very much for the review and valuable comments.
The amendment was done to the Materials & Methods section (lines: 182-184).
Reviewer 3 Report
Dear Authors!
I want to congratulate for the interesting study conducted which I could have supported unless a very serious flaws that I found.
Major remarks
1. The title does not reflect findings correctly. I believe you can not discuss the efficacy of alteplase based on just a mortality in a group with extremely high risk of mortality due to COVID19. Your data had to be supported by CTPA or echo-cardiography also.
What you did find is the trend to higher mortality in patients with RHT.
2. You did not clearly explain in Materials and methods the reason why patients in group 3 did not receive thrombolysis. That they refused seems slightly strange as the threat for their life was extremely high. If compare this group with group 2 we can see that those who refused were older, more often males, more often obese, had higher CRP. That may indicate another reason behind not using thrombolysis.
3. ESC doesn’t recommend to use thrombolysis for intermediate risk PE patients. But you included them and performed thrombolysis as I understand. What was the reason for that?
4. You must show how many patients in each group were of intermediate risk. This might have a significant impact on treatment outcomes. One can suppose that those who died had high risk PE and those who survived had intermediate risk PE.
Distribution of patients at different risks must be presented.
5. I strongly disagree with your decision not to include RHT and thrombolysis in a logistic regression. Those variables must be included.
6. You extracted data on 364 patients in ICU. Why did not you compare mortality rate in those with and without PE? I believe this is important in justification of clinical significance of PE and thrombolytic therapy.
None
Author Response
Dear Reviewer, thank you very much for the review and valuable comments.
Please find the answers to your questions below:
1. The title does not reflect findings correctly. I believe you can not discuss the efficacy of alteplase based on just a mortality in a group with extremely high risk of mortality due to COVID19.
It is changed
2. Your data had to be supported by CTPA or echo-cardiography also. What you did find is the trend to higher mortality in patients with RHT.
Please note, that as mentioned in Material & Methods section, all patients underwent ultrasound examination and CT-PA was done in 67/74 (90.5%) patients. In all cases, the diagnosis of thromboembolism has been confirmed.
3. You did not clearly explain in Materials and methods the reason why patients in group 3 did not receive thrombolysis. That they refused seems slightly strange as the threat for their life was extremely high. If compare this group with group 2 we can see that those who refused were older, more often males, more often obese, had higher CRP. That may indicate another reason behind not using thrombolysis.
An amendment was added in the in Material & Methods section (lines: 182-184). We added a comment to the Discussion section (lines: 336-339).
ESC doesn’t recommend to use thrombolysis for intermediate risk PE patients. But you included them and performed thrombolysis as I understand. What was the reason for that?
There is a misunderstanding concerning the phrase “patients with high-risk PE”. We used it in the same meaning as the ECS guidelines i.e. patients with confirmed PE and a high risk of PE-related death” and not patients with a high probability of PE. The inclusion criteria have been redrafted to make this clear (lines: 121, 124)
Consequently, our patients (with confirmed PE and a high risk of death due to PE) had indications for thrombolytic therapy.
4. You must show how many patients in each group were of intermediate risk. This might have a significant impact on treatment outcomes. One can suppose that those who died had high risk PE and those who survived had intermediate risk PE. Distribution of patients at different risks must be presented.
There is a misunderstanding resulting from our inaccuracy in the inclusion criteria. This should be based on the generally accepted guidelines, namely the ESC-ERS Guidelines for the diagnosis and management of acute pulmonary embolism [14]. According to Table 8: “Haemodynamic instability, combined with PE confirmation on CTPA and/or evidence of RV dysfunction on TTE, is sufficient to classify a patient into the high-risk PE category.”
Thus, all our patients fulfill the criteria of PE-related high risk of death, and the phrase “high-intermittent risk” should not be used concerning our population.
5. I strongly disagree with your decision not to include RHT and thrombolysis in a logistic regression. Those variables must be included.
Please note that both thrombolysis and RHT were the discriminating features between the investigated groups (patients with or without RHT and the ones with and without thrombolysis). Unfortunately, this is a retrospective study. Thus, a “head-to-head” comparison was not possible. Anyway, in those patients who underwent thrombolysis:
- surprisingly, RHT has no effect on the course of the disease (table 3),
- not the thrombolysis itself, but the delay in thrombolytic treatment was investigated and it was found significant for the outcome of the disease (tables 3 and 4).
This was discussed in the Discussion section and included in the Conclusions.
6. You extracted data on 364 patients in ICU. Why did not you compare mortality rate in those with and without PE? I believe this is important in justification of clinical significance of PE and thrombolytic therapy.
Indeed, critically ill COVID-19 patients with or without PE are distinct populations in respect to mortality. In our material, those without PE had mortality rate of 27.2% while in the study population mortality was 58.1% (data not included to the text). The study is aimed to estimate alteplase efficacy in pulmonary embolism thus we did not include other critically ill COVID-19 patients without indications for thrombolytic therapy.
Round 2
Reviewer 1 Report
Retrospective articles, in addition to evaluated information, need a very comprehensive introduction and discussion section with current information to be very valuable, especially as it is retrospective in nature and present certain unmet needs. In this sense, it is imperative to approach the management of COVID-19 based on the suggestions received.
Author Response
Dear Reviewer, we extended the Introduction section and added 8 references.
Reviewer 3 Report
Dear Authors!
Thank you for addressing my remarks. Now I'm Ok with your paper.
None
Author Response
Thank You very much.